# Optimized Spheroid Model of Pancreatic Cancer Demonstrates Influence of Macrophage–T Cell Interaction for Intratumoral T Cell Motility

**DOI:** 10.3390/cancers17010051

**Published:** 2024-12-27

**Authors:** Benedikt Slusny, Vanessa Zimmer, Elena Nasiri, Veronika Lutz, Magdalena Huber, Malte Buchholz, Thomas M. Gress, Katrin Roth, Christian Bauer

**Affiliations:** 1Department of Gastroenterology, Endocrinology, Infectious Diseases and Metabolism, University Hospital Marburg, 35043 Marburg, Germany; slusny@students.uni-marburg.de (B.S.); elena.nasiri@gmail.com (E.N.); malte.buchholz@staff.uni-marburg.de (M.B.); gress@med.uni-marburg.de (T.M.G.); 2Institute of Systems Immunology, Center for Tumor Biology and Immunology, Philipps University Marburg, 35043 Marburg, Germanymagdalena.huber@staff.uni-marburg.de (M.H.); 3Core Facility Cellular Imaging, Center for Tumor Biology and Immunology, Philipps University Marburg, 35043 Marburg, Germany; katrin.roth@imt.uni-marburg.de; 4Department of Gastroenterology, DonauIsar Klinikum Deggendorf, MedizinCampus Niederbayern, 94469 Deggendorf, Germany

**Keywords:** 3D spheroid model, pancreatic cancer, T cell motility, macrophages, IL-18, NLRP3

## Abstract

Three-dimensional spheroid models better reflect tumor biology compared to cell lines growing two-dimensionally in culture dishes. The 3R principle of replacement, reduction, and refinement of animal experiments has further enhanced interest in innovative 3D spheroid techniques. The present study establishes a triple cell spheroid model of pancreatic carcinoma that characterizes intratumoral T cell motility as a parameter of T cell effector function. Spheroid formation upon coculture of tumor cells, macrophages, and CTLs allowed intratumoral T cell migration analysis. Interaction between macrophages and highly mobile CTLs that recognized the model antigen OVA on pancreatic cancer cells was visualized, and parameters of T cell and macrophage motility were characterized. The presence of macrophages increased T cellular motility. Treatment of macrophages with NLRP3 inflammasome activator nigericin increased the velocity of intratumoral CTLs, but decreased CTL arrest on target cells and cytotoxic efficacy. Our work aims to establish T cell motility analysis in tumor spheroids as a readout parameter of T cellular effector function, contributing to ongoing efforts to refine animal experiments in cancer research.

## 1. Introduction

Despite advances in combination chemotherapy, pancreatic carcinoma is one of the deadliest cancer entities with a 5-year survival rate of 10%. Intratumoral CD8^+^ T cells exert cytotoxic effector functions and their tumor infiltration correlates with improved survival in patients with pancreatic carcinoma [1]. However, antitumoral T cell responses are limited in efficacy due to low immunogenicity of pancreatic cancer because of a highly immunosuppressive tumor milieu. Immunosuppressive cell populations within the tumor stroma include macrophages [2], myeloid-derived suppressor cells (MDSCs) [3], and regulatory T cells (Tregs) [4]. Furthermore, pancreatic cancer cells are characterized by exceptionally low antigenicity, resulting from low mutational burden and absence of neoantigens [5]. However, subtypes of pancreatic cancer, displaying mismatch repair deficiency, are susceptible to checkpoint inhibition [6]. Additionally, genomic profiling has identified subtypes that are more susceptible to immunotherapy than others [7]. This demonstrates that a better understanding of cellular and molecular interactions between T cells and other components of the tumor stroma might pave the way to T-cell-based immunotherapy of pancreatic cancer.

Due to the complexity of cellular interactions in tumorigenesis and antitumor immune responses, animal experiments are considered irreplaceable. However, adherence to the 3R principle of replacement, reduction, and refinement of animal experimentation has resulted in increasing interest in alternative methods of investigating interplay of cellular components of tumor biology. Spheroid models and primary tumor organoids are superior to in vitro assays using cell cultures that grow two-dimensionally in culture dishes. The present study develops a model for motility analysis of T cells, based on spheroid formation with coculture of three different cell entities (tumor cells, macrophages, and T cells). The study aims to establish T cell motility as a readout parameter of intratumoral interaction between T cells and macrophages. We hypothesize that molecular interventions such as treatment of macrophages with lipopolysaccharide (LPS) and nigericin for NLRP3 activation significantly affect T cell migration within spheroids. Our own work has demonstrated that T cellular IL-18 receptor (IL-18R) signaling influences intratumoral migration patterns of CTLs [8]. Activated NLRP3 inflammasome of macrophages is a major producer of active IL-18. Here, the role of LPS-/nigericin-induced pathways in myeloid cells for CTL motility was investigated in a refined spheroid model of pancreatic cancer.

Tumor-associated macrophages play a central role in shaping the immune microenvironment. Traditionally, macrophages are believed to be polarized toward an M1 or M2 phenotype. This concept was developed based on in vitro data and had a striking similarity to the already established Th1/Th2 paradigm for T cells [9]. The original concept suggested a dichotomy between M1 macrophages that kill infectious organisms or tumor cells and M2 macrophages that repair and heal tissue damage [10]. Within this framework, M1 macrophages were found to kill cancer cells and M2 macrophages to promote cancer [11]. Over the years, many more macrophage subtypes have been described, and it is now clear that the differentiation of monocytes into macrophages is a complex process resulting from cytokine stimulation and contact-dependent interaction with other cell types [12]. However, to compare results to reports in the literature, standardized stimuli are helpful to induce characteristic macrophage phenotypes. IL-4 stimulation is considered to polarize macrophages toward an M2 phenotype [13].

Motility of cells is a dynamic process characterized by the reorganization of the cytoskeleton. In case of T cell migration, this is associated with polarization of cellular morphology [14]. T cells extend protrusions in the direction of migration. This is linked to a network of filamentous actin with the leading edge being particularly sensitive to the engagement of T cell antigen and chemokine receptors [15]. Cytotoxic T cells (CTLs) use LFA-1–ICAM interactions to move through junctions between target cells [16]. Motility of CTLs is closely linked to their cytotoxic effector function [17]. Therefore, migration toward target cells is very important, and this process is regulated and guided by chemokines such as CXCL12, CXCL10, and CCL2, produced by tumor cells, tumor-associated macrophages, and fibroblasts [18]. When CTLs interact with their target cells, they form an immunological synapse, followed by the release of perforin and granzyme and the secretion of effector cytokines such as TNF and IFN-γ [19]. Interaction is characterized by MHC-I-mediated contact with the cognate antigen. Therefore, CTL motility directly affects the frequency of cytotoxic interactions. Importantly, intratumoral migration per se is undirected, often described as a “random walk” [20]. CTLs perform migration arrest phases, which are believed to result from engagement with target cells. These target cells can be either tumor cells or antigen-presenting cells (APCs), such as macrophages or dendritic cells (DCs).

The role of these APCs remains a matter of debate. Macrophages have been described to impede CTLs from reaching tumor cells and limit antitumor therapy [21]. Our own work has shown that myeloid cells act as bridging cells to transmit suppressive signals from Tregs to CTLs in an antigen-dependent manner [20]. Tregs conditioned DCs to reduce their costimulatory activity, resulting in augmented coexpression of PD-1 and TIM-3 by CTLs, suggesting enhanced T cell exhaustion. Motility analysis demonstrated specific Treg and CTL migration patterns in a dorsal skinfold chamber model.

However, it remained unclear if and how APCs per se promoted changes to the effector function and motility patterns of CTLs. Here, we suggest that the polarization and activation state of macrophages defines their role as platform cells for integrating activating and suppressive signals to CTLs. Among many molecular pathways that influence the production of cytokines and the expression of costimulatory and coinhibitory molecules, the NLRP3 inflammasome and NLRP3-dependent cytokines IL-1β and IL-18 are pivotal for the fate of CTL effector function [19]. This is also reflected by significant changes to motility when CTLs are deficient for IL-1 receptor (IL-1R) or IL-18 receptor (IL-18R) signaling [8]. IL-18R-deficient CTLs (and, to a lesser degree, IL-1R-deficient T cells) have been shown to have enhanced velocity and more frequent arrests when moving through antigen-rich tumor environments. This was demonstrated in a dorsal-skinfold chamber model and a tumor spheroid model, employing a Matrigel matrix. Results of the skinfold chamber model and the spheroid model were remarkably similar, indicating that tumor spheroids are well suited to investigate motility and migratory patterns of CTLs.

In this paper, we have extended our spheroid model by the addition of macrophages. Here, motility analysis of macrophages and changes in motility of CTLs as a result of CTL–macrophage interaction were investigated. To define the role of macrophages as platform cells for integrating antigen-dependent motility signals, macrophages were polarized toward an M2 phenotype, and then treated with LPS and nigericin or not in order to activate the NLRP3 inflammasome. Imaging of macrophages and T cells in tumor spheroids demonstrated consistent patterns of migration. Importantly, increased velocity of T cells upon incubation of macrophages with nigericin did not result in an increase in T cell arrest phases and cytotoxicity. To the contrary, nigericin treatment was associated with decreased arrest and slightly reduced spheroid rejection. These results point toward an immunoregulatory role of nigericin-induced pathways such as NLRP3 activation in tumor immunology, supporting our own data on a role for IL-18 in limiting effector function of T cells under specific conditions of intratumoral antigen recognition [19]. This indicates that an optimized triple-cell-type spheroid model of pancreatic cancer, as developed here, can consistently serve as a model to study regulation of T cell motility and effector function. Sophisticated tracking of infiltrating versus peripheral macrophages and CTLs is feasible in this model, allows intervention on a molecular level (such as NLRP3 activation), and establishes T cell motility as a readout parameter of tumor biology studies.

## 2. Materials and Methods

### 2.1. Cell Lines

Murine pancreatic ductal adenocarcinoma (PDAC) cell line Panc02 (RRID:CVCL_D627), which was originally derived from a methylcholanthrene-induced pancreatic ductal adenocarcinoma in C57BL/6 mice [22], was stably transfected to express ovalbumin (OVA) as described before (PancOVA) [23]. Cell lines Panc02 and PancOVA were gifts from Max Schnurr (University of Munich, Munich, Germany). Using a lentiviral transfection system, cells were stably transduced with plasmid pCS-H2B-Cerulean (#53748, Addgene, Watertown, NY, USA) [8]. Cell lines were routinely tested for Mycoplasma contamination. Tumor cells were cultured in DMEM (Gibco Thermo Fisher Scientific, Waltham, NY, USA, catalog no. 41965062) supplemented with 10% FCS (Capricorn Scientific, Ebsdorfergrund, Germany, catalog no. FBS-11A) and 1% Pen/Strep (Gibco, catalog no. 15070-063). A total of 500 mg/L G418 (Sigma-Aldrich, St. Louis, MI, USA, catalog no. G8168-10ML) was added to PancOVA cells for positive selection. Cells were used for experiments until passage 20.

### 2.2. Mice

OT-I transgenic T cell receptor (RRID: IMSR_JAX:003831) and TgVenus (RRID: IMSR_ JAX:011107) mice were obtained from Jackson Laboratory. OT-I mice were mated with B6.129P2-*Il18rltm1Aki*/J (RRID: IMSR_JAX:004131) mice to generate a strain with transgenic T cell receptor and IL-18R deficiency. All mice had a C57BL/6J background. A total of thirteen TgVenus mice were sacrificed to generate bone-marrow-derived macrophages (BMDMs). Furthermore, nine OT-I mice and seven *Il18r^−/−^* OT-I mice were sacrificed to generate CTLs.

### 2.3. Isolation of Murine Bone-Marrow-Derived Macrophages and Differentiation into M2-like Macrophages

TgVenus-positive mice were used to generate BMDMs and sacrificed by cervical dislocation. Both femurs were isolated, and bones were rinsed with HBSS. The bone-marrow-derived cells were flushed through a 70 µm cell strainer (Miltenyi Biotec, Bergisch Gladbach, Germany, catalog no. 130-041-407) and then lysed for 3 min to remove erythrocytes. After centrifugation, the BMDMs were adjusted to 1 × 10^6^ cells/mL and seeded in a 24-well plate with 50 ng/mL murine recombinant M-CSF-1 (rm-M-CSF-1; Peprotech Thermo Fisher Scientific, Waltham, MA, USA, catalog no. 315-02) in RPMI 1640 medium (Gibco, catalog no. 21875-091) supplemented with 10% FCS, 3% HEPES (Gibco, catalog no. 15360-056), 2% Pen/Strep, 2% NaPyr (Gibco, catalog no. 11360-070), and 0.1% 2-mercaptoethanol (Gibco, catalog no. 31350010). After four days of differentiation, cells were treated with 20 ng/mL murine rm-IL-4 (Peprotech, catalog no. 214-14) for 24 h to polarize them into M2-like macrophages. After polarization of macrophages, cells were treated with 200 ng/mL LPS for 2 h and 10 µM nigericin for 1 h to activate the NLRP3 inflammasome. Cells were stained for F4/80, CD11b, CD80, CD206, and MHCII expression for 30 min in PBS (Gibco, catalog no. 14190-169) with 3% FCS at 4 °C in the dark. Flow cytometry was performed on a CytoFLEX LX6 (Beckman Coulter, Brea, CA, USA) and analyzed using the FlowJo software (BD, Franklin Lakes, NJ, USA, RRID:SCIR_008520, version 10.10).

### 2.4. Isolation and Differentiation of Murine Cytotoxic CD8^+^ T Cells

Murine T cells were isolated and differentiated as described before [8,19]. CTLs were generated from the lymph nodes and spleen of wild-type (WT) and *Il18r^−/−^* OT-I mice. After cells were mashed through a 30 µm cell strainer and lysed for 5 min, cells were adjusted to 4 × 10^7^ cells/mL and activated for 1 h in the presence of 5 µg of OVA peptide SIINFEKL (Invitrogen Thermo Fisher Scientific, Waltham, MA, USA, catalog no. code vac-sin). The cells were cultured for two days in complete RPMI 1640 medium supplemented with 10 ng/mL murine rm-IL-12 (Peprotech, catalog no. 210-12), followed by three days of cultivation with 20 ng/mL murine rm-IL-2 (Peprotech, catalog no. 212-12). Cells were analyzed via flow cytometry for the expression of CD8, CD25, CD44, CD69, and CD62L.

### 2.5. CellTracker™ Staining of Murine CTLs

CTLs were collected, and the cell number was adjusted to 2 × 10^6^ cells/mL. Cells were stained in PBS with 1 µM CellTracker™ DeepRed (Invitrogen, catalog no. C34565) for 45 min at 37 °C and 5% CO_2_. Afterwards, T cells were adjusted to 1.2 × 10^5^ per 300 µL RPMI 1640 medium with 10% FCS, 3% HEPES, 2% Pen/Strep, 2% NaPyr, and 0.1% 2-mercaptoethanol.

### 2.6. Generation of Spheroids and Coculture

To generate three-dimensional (3D) spheroids that resemble in vivo tumors morphologically and in terms of cellular composition, Corning^®^ Matrigel (Corning, Corning, NY, USA) was used to mimic the extracellular matrix of the tumor environment. In total, 6 × 10^4^ PancOVA H2B Cerulean cells were gently mixed in 45 µL DMEM medium with 15 µL Corning^®^ Matrigel on ice and carefully pipetted into FCS-coated Ibidi™ TruLive 3D dishes. After 15–20 min of incubation at 37 °C and 5% CO_2_, 300 µL complete RPMI 1640 medium was added. For the coculture with BMDMs, previously polarized macrophages with activated or not activated NLRP3 inflammasome were adjusted to 8 × 10^3^ cells and resuspended with the tumor cells in 45 µL DMEM medium before adding 15 µL Corning^®^ Matrigel. After 24 h of myeloid–spheroid coculture, 1.2 × 10^5^ WT or *Il18r^−/−^* CTLs were carefully added to the spheroids on top of the dish. TruLive 3D dishes were placed in the incubation chamber of the light sheet microscope TruLive 3D (Luxendo-Bruker, Heidelberg, Germany). Microscopic imaging started 4 h after adding CTLs to the spheroid–macrophage coculture for the early time point and 18 h for the late time point.

### 2.7. Staining of Caspase-3/7 Activity

To validate tumor cell killing in the spheroid coculture, a detection reagent for the activity of caspase-3/7 was added after the late time point and incubated for 60 min before initiating microscopic imaging. The detection reagent of the CellEvent™ Caspase-3/7 kit (Invitrogen, catalog no. C10430) consists of a nucleic acid binding dye conjugated to a four amino acid peptide (DEVD). During apoptosis, caspases-3 and -7 are activated, cleaving the DEVD peptide, thereby enabling the dye to bind DNA and resulting in fluorescence.

### 2.8. Analysis

Imaging data were analyzed using Imaris software (Bitplane, Belfast, UK, version 9.9.0 and 10.2). The surface algorithm with a high smooth factor was used to achieve a surface covering the whole spheroid and macrophages. Then, CTL and macrophage channel masks were set to discriminate between CTLs and macrophages inside the spheroid (infiltrating) and outside the spheroid (peripheral). CTLs (infiltrating, peripheral, and overall) were analyzed using the spot algorithm. If necessary, a drift correction was implemented for both surfaces and spots. The following parameters could be exported as Excel files from Imaris: spheroid volume, number of surfaces, number of infiltrating and peripheral CTLs, number of infiltrating and peripheral macrophages, average speed, straightness, track displacement, track duration, instantaneous velocity, track length, and distance from surface to spot. Using RStudio (2023.09.1+494), the arrest durations and arrest coefficients were calculated based on instantaneous velocity data, as described before [8].

### 2.9. Statistics

GraphPad Prism v10 (RRID:SCR_002798) and RStudio (2023.09.1+494) were used for statistical analysis. Datapoints represent biological replicates unless stated otherwise. If necessary, Grubbs’s test for detecting significant outliers was performed (alpha = 0.05). The unpaired, two-tailed *t*-test was used to compare groups of two independent samples. If the variances differed significantly, Welch’s correction was performed. For comparison of more than two groups, an ordinary one-way ANOVA followed by Tukey’s multiple comparisons was used, in which each group was compared to every other group. Statistical significances with *p* values were defined as * *p ≤* 0.05, ** *p* < 0.01, and *** *p* < 0.001. Data are represented as bar graphs (mean ± standard deviation (SD)).

## 3. Results

### 3.1. Visualization of Macrophage–T Cell Interaction in a Spheroid Model of Pancreatic Cancer

Building on previous work describing a spheroid model of pancreatic cancer T cell infiltration [8], we established spheroid formation in the presence of in vitro-generated macrophages (Figure 1A). Matrigel-based spheroids were derived from PancOVA H2B Cerulean cells, SIINFEKL-reactive OT-I CD8^+^ CTLs, and CD206^+^ macrophages. CTLs recognizing the OVA-derived peptide SIINFEKL originated from wild-type (WT) or *Il18r*^−/−^ OT-I mice and were stained with CellTracker™ Deep Red. No differences between WT and *Il18r*^−/−^ OT-I effector T cells were found upon phenotypic FACS analysis of activation markers CD25, CD62L, CD44, or CD69 as well as intracellular TNF and IFN-γ (Appendix A). Macrophages were derived from the bone marrow of TgVenus-positive mice, allowing later visualization of TgVenus by using the light sheet microscope TruLive 3D. Macrophages were polarized into M2-like CD206^+^ cells by treatment with 20 ng/mL murine rm-IL-4 (Peprotech) and either activated with LPS and nigericin or left not activated before spheroid formation began. Treatment with LPS and nigericin did not change the phenotype of macrophages upon staining of surface markers CD80, CD206, and MHC-II (Appendix A). After 24 h of incubation, CD8^+^ CellTracker DeepRed-stained CTLs were added to the spheroids. Microscopic imaging was performed after 4 h (early time point), overnight, and 18 h (late time point), respectively. Imaging analysis was performed using Imaris software. This approach resulted in 3D spheroids consisting of H2B Cerulean-positive tumor cells, CellTracker DeepRed-stained CD8^+^ CTLs, and TgVenus-positive CD206^+^ macrophages (Figure 1B, Appendix A). Imaris-based analysis enabled the distinction between peripheral CTLs (green), located strictly outside the spheroid, and infiltrating CTLs (red). Upon visualization, different colors were assigned to these two populations, and track analysis was performed separately. Similarly, CD206^+^ macrophages were defined and subsequently analyzed as peripheral (yellow) and infiltrating (purple) macrophages (Figure 1C, Appendix A). As for CTLs, tracking analysis demonstrated the high mobility of these infiltrating CD206^+^ macrophages within the tumor spheroid (Figure 1D). Based on the Imaris spot algorithm, the percentages of infiltrating antigen-specific CTLs among all CTLs in PancOVA spheroids were calculated (Figure 1E, *n* = 3–7). After 4 h (early, left), 25.3 ± 14.37% of WT T cells were found within the spheroid volume. Infiltration had increased to 30.22 ± 4.48% after 18 h (late, right). There was a trend toward increased infiltration of *Il18r^−/−^* CTLs, corroborating our earlier report on CTL motility [8]. Interestingly, infiltration was slightly increased in the presence of macrophages. NLRP3 activation status of these macrophages did not significantly influence the percentage of infiltrating CTLs. However, the highest percentage of CTLs within the spheroid was found for *Il18r^−/−^* CTLs in the presence of nigericin-treated macrophages after 18 h (Figure 1E, right, dark orange bar). Similarly, the number of peripheral and spheroid-infiltrating CD206^+^ macrophages after 4 h and 18 h of coculture was determined using the surface algorithm of Imaris software (Figure 1F, *n* = 6–7). The total number of untreated macrophages (identified by TgVenus positivity) within the spheroids increased by 115% from 4 h to 18 h (5.55 ± 3.54 at 4 h versus 11.95 ± 5.04 at 18 h, light blue bars), indicating directed migration of macrophages into spheroids over time. Furthermore, these data suggest the viability of macrophages throughout the time frame of the experiment. Importantly, nigericin treatment resulted in a slight increase in early macrophage infiltration into spheroids. The difference between infiltrating and peripheral macrophages was significant for non-activated macrophages at the early time point but not for macrophages treated with nigericin in order to activate NLRP3 (Figure 1F left, dark blue versus light blue bars). In summary, a macrophage–T cell interaction model within PancOVA spheroids was established. Within this model, nigericin treatment for NLRP3 activation had modest but partly significant effects on the infiltration behavior of T cells and macrophages. Using this spheroid model of pancreatic carcinoma, intratumoral motility of antigen-specific T cells was investigated.

### 3.2. Nigericin Treatment of Intratumoral CD206^+^ Macrophages Increases CTL Motility

Imaris spot algorithm analysis of migratory tracks, visualized by light sheet microscopy, was employed to investigate the influence of incubation of macrophages with LPS and nigericin on migratory parameters of intratumoral WT and *Il18r^−/−^* CD8^+^ CTLs. Videos of 30 min length were recorded at 4 and 18 h after the start of coculturing PancOVA spheroids with ex vivo generated OT-I CD8^+^ CTLs. Average speed was determined as an overall parameter of T cell motility. Concordant with previous results [8], *Il18r^−/−^* CD8^+^ CTLs were faster than WT CTLs. Notably, macrophage coculture significantly increased the average speed of CTLs, but, only when macrophages had been treated with LPS and nigericin for NLRP3 activation (Figure 2A). Coculture of *Il18r^−/−^* CTLs with nigericin-treated macrophages did not further increase CTL motility. However, upon comparison of CTLs in the presence of activated versus non-activated macrophages, the presence of nigericin-treated macrophages increased the average speed of *Il18r^−/−^* CTLs. This may indicate that the effects of LPS and nigericin were not solely based on the secretion of NLRP3-mediated cytokines IL-1β and IL-18 into the culture chamber, since T-cell-intrinsic direct effects of NLRP3-mediated cytokines should have been absent in *Il18r*^−/−^ CTLs.

Similarly, another parameter of T cell motility, track displacement, was also increased in both WT and *Il18r^−/−^* CTLs by coculture with activated macrophages (Figure 2B). Track displacement describes the vectorial distance between the start and end of a CTL track. Additionally, calculating the total length of displacement within the track, shown as track length (Appendix A), revealed a tendency toward increased distance covered by the T cells during coculture with macrophages possessing an activated NLRP3 inflammasome. Interestingly, the effects of IL-18R deficiency and activation with nigericin on both parameters were abrogated at the late time point, after 18 h of coculture. Straightness indicates directional (index close to 1) and undefined (index close to 0) movement of cells. It is calculated by dividing the track displacement of a CTL by its track length. No differences in the migration straightness of CTLs in coculture with non-activated and activated macrophages were found (Appendix A). In our dataset, this parameter was around 0.4 to 0.5, suggesting that all CTLs performed a random-walk-like motility pattern, irrespective of whether activated or non-activated macrophages were around. In summary, intratumoral activation of macrophages with LPS/nigericin increased velocity and traveled distance of T cells, but did not significantly affect directionality of their movement.

Furthermore, motility analysis was performed on spheroid-infiltrating and peripheral macrophages. Average speed (Figure 2C) and track displacement (Figure 2D) were largely independent of location and NLRP3 activation status, both at the early and late time points. Upon statistical analysis, the average speed of WT infiltrating macrophages was found to be significantly slower than speed of WT peripheral macrophages. This effect was absent in the presence of macrophages incubated with LPS and nigericin. Here, macrophages moved faster, which correlated with the finding of increased intraspheroidal macrophage infiltration after coculture with activated macrophages (Figure 1E). However, the overall effect strength was small, with the movement pattern of macrophages being rather uniform throughout all subgroups. Analysis of track length and straightness related to the findings regarding macrophages’ average speed and track displacement (Appendix A). In summary, performing track analysis of T cells from *n* = 99 spheroids, each consisting of about 114 individual CTL track data (minimum 6 tracks, maximum 767 tracks, with a mean of 113.5 recorded track data), our findings demonstrate that activation of intratumoral CD206^+^ macrophages with LPS and nigericin increased the motility of antigen-specific CTLs within PancOVA spheroids.

In summary, nigericin treatment of macrophages resulted in increasing velocity of CTLs; however, data on track displacement, track length, and straightness (parameters that are linked to directed migration) indicate that these high-velocity CTLs are not necessarily more efficient. CTLs that were cocultured with nigericin-stimulated macrophages moved in rather random directions through the imaging field, in which plenty of cognate antigen was expressed. These data can be put into a broader context by investigating the cytotoxic effector function of those CTLs. Here, nigericin treatment decreased macrophage-associated increases in T cell arrest and rejection capacity (see below, Figure 3 and Figure 4).

### 3.3. Pre-Treatment with LPS and Nigericin Abrogates Macrophage-Induced Increase in T Cell Arrests in PancOVA Spheroids

CTL migration in an environment that is devoid of the cognate antigen is undirected and continuous [24,25]. Upon antigen recognition, the fast-paced scanning of CTLs is converted into a slower motility pattern. Phases of slow motility are termed arrest phases and are believed to represent the engagement of adhesion-promoting and costimulatory molecules that result in the release of cytotoxic granules. The arrest coefficient, defined as the fraction of time during which an individual cell is immobilized, is, therefore, an insightful parameter to characterize the functional state of effector T cells (Figure 3, Appendix A). Recently, we described that OT-I CTLs demonstrate an undirected movement pattern in spheroids that lack expression of the model antigen OVA. In contrast, movement in PancOVA spheroids was directed and characterized by frequent arrests, defined as a fall of velocity below a threshold of 2 µm/min [8]. Here, the measurement of CTL track behavior was extended to settings in the presence of nigericin-stimulated and non-stimulated macrophages. Importantly, as in previous experiments [8], *Il18r^−/−^* CTLs demonstrated more frequent arrests than WT CTLs (Figure 3A,C, after 4 h of coculture). Late measurements demonstrated an overall higher arrest coefficient throughout all groups, possibly associated with dissolution of the spheroid and dispersed presence of antigen within the migratory field (Figure 3C,D and Figure 4D). This could also explain the additional peak in higher arrest coefficients at the late time point compared to the early time point (Appendix A). Interestingly, the presence of macrophages increased the arrest coefficient of CTLs in the early phase (Figure 3C, light gray bar). However, nigericin treatment of macrophages resulted in a slight decrease in the arrest coefficient (dark gray), also found for early infiltrating *Il18r^−/−^* cells (dark orange). The arrest coefficient correlated well with arrest duration of CTLs in the early phase (Figure 3E,F). These data have to be correlated with cytotoxicity data presented in Figure 4, as seen below, and show quite remarkable homology of patterns when comparing groups in the presence and absence of nigericin stimulation.

### 3.4. CTL Cytotoxicity Is Increased by Coculture with Intraspheroidal Macrophages

To investigate the effects of myeloid cell NLRP3 activation on CTL-induced tumor rejection, PancOVA spheroids were cultured with combinations of macrophages (either treated or not treated with LPS and nigericin) and OVA-reactive OT-I WT or *Il18r^−/−^* CD8^+^ effector CTLs (Figure 4A,B). Tumor size progression was observed overnight, and fold change in spheroid volume was measured (Figure 4C). As another parameter of tumor rejection, fold change in spheroid fragments was calculated (Figure 4D, Appendix A). Spheroid rejection was increased, indicated by decreased spheroid volume, when CTLs and macrophages were cocultured with PancOVA. Interestingly, activation of macrophages through nigericin resulted in a very slight loss of rejection capacity for WT and *Il18r^−/−^* CTLs. As mentioned above, this is paralleled by a slight decrease in the arrest coefficient when CTLs were cocultured with treated macrophages (compared to non-treated ones). This phenomenon might be based on increased PD-L1 expression on the surface of macrophages through NLRP3 activation and subsequent suppressive effects on CTL cytotoxicity.

Finally, to investigate the interplay of macrophages and CTLs during cytotoxic rejection, 3D depictions of PancOVA spheroids were calculated based on light sheet microscope TruLive 3D data. Caspase-3 and -7 staining confirmed cytotoxic activity of OT-I CD8^+^ CTLs in PancOVA spheroids (Figure 4E, Appendix A). To validate the role of CTLs in apoptosis induction, the mean distance of CTLs to those cells that stayed alive and those that entered a fluorescent state, indicating caspase-3 and -7 positivity, was calculated. Here, apoptotic tumor cells were significantly closer to CTLs than those cells that stayed alive and healthy (and did not stain positive for caspase-3 and -7).

To sum up, our data indicate that the NLRP3 activation state in macrophages governs CTL motility and cytotoxicity. More specifically, nigericin treatment of macrophages resulted in increased velocity of cocultured CTLs within the organoid. However, this did not translate into increased CTL cytotoxicity. To the contrary, although the presence of macrophages increased the arrest coefficient as a parameter of cytotoxicity, incubation with LPS and nigericin of those macrophages resulted in a slight decrease in CTL arrests on target cells.

## 4. Discussion

In this paper, 3D pancreatic cancer spheroids were established as a tool to investigate intratumoral T cell effector function by analyzing T cell motility. Extending previous work on tumor spheroids [8], a triple cell spheroid model of pancreatic cancer was generated by seeding three different, ex vivo generated cell types: tumor cells, CTLs, and macrophages. Time-lapse imaging of spheroids allowed characterization of multiple endpoints of T cell motility, such as average velocity, track length, straightness, and arrest coefficient. Mechanistically, this paper investigates the role of nigericin treatment of macrophages for CTL motility and tumor rejection.

As a potassium ionophore, nigericin is a well-described activator of the NLRP3 inflammasome, often used as a positive control in experiments on NLRP3 activation [26]. Activation of NLRP3 requires a priming signal, e.g., by LPS stimulation (step 1), resulting in NF-kappaB-driven upregulation of NLRP3, pro-IL-1β, and pro-IL-18 [27]. Signal 2, e.g., through nigericin, then induces the assembly of the NLRP3 inflammasome, resulting in the release of IL-18 and IL-1β. In this paper, macrophages were treated with LPS and nigericin in order to investigate the involvement of NLRP3 activation status of intraspheroidal macrophages for CTL motility. However, as a caveat, nigericin has other targets than the NLRP3 inflammasome, such as inhibition of the Wnt/β-catenin signaling pathway [28]. Nigericin promotes NLRP3-independent mechanisms in macrophages [29], e.g., through cathepsin B activation, resulting in caspase-1 activation and IL-18 maturation [30]. Accordingly, further experiments with NLRP3 inhibitors like MCC950 are warranted to exactly delineate strictly NLRP3-mediated effects. However, in this paper, nigericin serves as an established and reliable activator of the NLRP3 inflammasome in macrophages, and we assume that effects on T cell motility observed after coculture with LPS-/nigericin-treated macrophages are predominantly mediated by NLRP3 activation. Accordingly, our data indicate that the activation state of myeloid cells, induced by LPS and nigericin treatment, directs CTL motility and effector function.

As found in many models investigating NLRP3 effects [31,32], NLRP3 activation had no straightforward, unidirectional impact on antitumor immunity. Instead, treatment of macrophages with LPS/nigericin resulted in the augmentation of some parameters (T cell velocity), but a decrease in others (T cell arrest and cytotoxic efficacy). Our results strengthen the notion that APCs in the tumor stroma function as platform cells for the integration of immunosuppressive and immunoactivating signals. NLRP3 plays a vital role in defining the phenotype and function of intratumoral macrophages, resulting in secondary effects on the motility and function of intratumoral T cells.

Inflammasomes are large protein complexes that control caspase-1-mediated cleavage of the precursor molecules of cytokines IL-1β and IL-18, as well as Gasdermin D (GSDMD) [33]. Inflammasomes have been reported to have both anticancer effects and protumorigenic functions. Carcinogenesis is promoted through chronic activation of the NF-κB signaling pathway, which is activated in the priming step of NLRP3 inflammasome activation [34,35]. Furthermore, biologically active IL-1β is required for tumor invasiveness and angiogenesis [36]. On the other hand, NLRP3 mediates potent antitumor effects. The inflammasome NLRC4 and caspase-1 inhibit the formation of malignant colonic epithelial cells [37]. Crosstalk between NLRP3 and T cells has been suggested by demonstrating that activation of DC through ATP leakage from dead cells can result in production of IL-1β and IL-18, which in turn triggers secretion of IFN-γ from CD8^+^ T cells [34]. Our work found that NLRP3 is necessary for intratumoral T cell exhaustion, as IL-18 and IL-1 receptor deficiency resulted in the disinhibition of exhausted T cells [19]. These *Il18r^−/−^* and *Il1r*^−/−^ T cells demonstrated significant differences in motility and intratumoral interaction with target cells. Consistently, short-term effects of NLRP3 activation and secretion of IL-1β and IL-18 were stimulatory toward T cells. However, prolonged exposure of T cells to IL-1β and/or IL-18 resulted in suppressive effects on T cells and was detrimental to the cancer-bearing organism.

In this paper, *Il18r^−/−^* T cells were used to delineate T-cell-intrinsic effects of NLRP3-derived cytokine IL-18 from other effects that do not depend on NLRP3-/caspase-1-mediated release of cytokines IL-1β or IL-18. It was found that many effects on T cell motility mediated by nigericin-treated macrophages (such as average speed, but also track displacement and track length) were preserved in IL-18R-deficient T cells. This indicates that “non-IL-18” factors are (at least partially) responsible for the augmented velocity of T cells cocultured with nigericin-treated macrophages. This might be a not yet specified effect of IL-1R signaling on T cell motility or—more likely—an indirect effect that is not mediated by T-cell-intrinsic mechanisms, but by induction or suppression of coinhibitory and costimulatory molecules on other cell types. Possible candidates include CD80, CD86, and PD-L1 on myeloid cells. Concordantly, we observed ample interaction between macrophages and CTLs in our spheroid model.

A reciprocal regulatory relationship between inflammasomes and PD-1/PD-L1 signaling in tumorigenesis has been suggested [33]. PD-1 is a receptor with two ligands, PD-L1 and PD-L2. Signaling through PD-1 results in reduced T cell activity with effects on IL-2 secretion, T cell proliferation, and promotion of Tregs through metabolic reprogramming [38,39]. PD-1-based checkpoint inhibition has failed as monotherapy in many solid cancer entities, including pancreatic carcinoma, due to low PD-L1 expression, low tumor neoantigen expression, adverse epigenetic changes, and an immunosuppressive tumor microenvironment. The presence of M2 macrophages and MDSCs has been shown to render tumors less susceptible to PD-1 blockade [40]. NLRP3 is involved in these mechanisms.

Pharmacological inhibition of NLRP3 with small molecule MCC950 reduced PD-L1 expression [41]. The Dinarello group reported that activation of NLRP3 dictated a response to anti-PD-1 therapy in a breast cancer model and suggested that NLRP3 is a key driver of immune suppression [42]. In lymphoma, the NLRP3 inflammasome was found to upregulate PD-L1 expression, resulting in T cell exhaustion. Notably, the interaction between inflammasomes and PD-L1 plays a role in pancreatic cancer, too. DNA damage in PDAC cells leads to AIM2 inflammasome activation and—via the release of high mobility group protein 1 (HMGB1)—to the expression of PD-L1 [43]. Other checkpoint molecules, such as LAG-3 and IDO, can also be induced by inflammasome activation [41,44]. Interestingly, interaction between PD-1/PD-L1 and NLRP3 is reciprocal. Blockade of PD-1 on T cells triggered NLRP3 inflammasome activation through the STAT3 signaling pathway [45].

Other molecules and other cell types are involved, too. IL-18 enhanced the expression of CD80, CD86, HLA-DR, and HLA-DQ on NK cells, suggesting that IL-18 conferred NK cells an APC-like phenotype [46]. These APC-like NK cells then efficiently killed tumor cells. Fibroblasts might be necessary, too, as treatment of cancer-associated fibroblasts with IL-1β significantly enhanced their ability to inhibit CD8^+^ T cell function and proliferation [47], which was mediated by upregulation of PD-L1/2. IL-1β is known to help in maintaining the immunosuppressive function of MDSCs [48]. Our own data showed that reduced expression of CD80 and CD86 on DCs was associated with increased T cell exhaustion. This was correlated with migration pattern analysis of CTLs and Tregs in a dorsal skinfold chamber model designed for motility analysis [20]. Fife et al. reported that PD-1/PD-L1 interaction promotes tolerance by blocking the T-cell-receptor-induced stop signal [49].

It is tempting to speculate that NLRP3-induced increased expression of coinhibitory molecules such as PD-L1 on intraspheroidal macrophages and tumor cells is causing the slight decrease in spheroid rejection that was observed in our spheroid model. However, this has to be investigated in future studies that are not primarily devoted to motility analysis of intratumoral T cells and macrophages.

In our hands, pre-treatment of intratumoral macrophages with LPS and nigericin slightly decreased the arrest coefficient, which is concordant with the findings of Fife et al., and points toward a possible involvement of PD-1/PD-L1 interaction to our findings. The arrest coefficient is the most reliable parameter in linking T cell motility to cytotoxic efficacy, as it determines those phases in which a T cell is immobilized on a target cell [20]. This can either happen with macrophages when immunosuppressive and immunoactivating signals are exchanged or with tumor target cells when cytotoxic granules are released. Therefore, an increase in velocity and a decrease in arrest coefficient due to nigericin treatment is concordant with the slightly reduced cytotoxic efficacy of T cells toward our spheroids. This was again found in both WT and IL-18R-deficient T cell conditions, further strengthening the hypothesis of a non-T-cell-intrinsic mechanism that renders T cells faster but less effective.

Multiple studies found evidence that T cell motility is associated with tumor progression [24,50]. Checkpoint blockade by anti-CTLA-4 antibodies increased velocities of antigen-specific T cells in tumor-draining lymph nodes [51]. Multiphoton imaging demonstrated the contribution of stromal compartments and collagen fibers for T cell migration [50]. Based on these data, we decided to investigate T cell motility in two separate compartments, parenchyma-infiltrating and peripheral T cells. In a previous paper, we could describe T-cell-intrinsic effects of NLRP3-mediated cytokine signaling [8]. This was linked to significant changes in the differential expression of motility-associated target genes, such as Coro2a. However, these intrinsic effects were found in a spheroid model that consisted only of T cells and tumor cells. This is an important caveat, as motility, especially cytotoxic contact time, differs significantly between tumor immunology and, e.g., viral infection models [52]. T cells were markedly slower in the tumor microenvironment, and this might be due to the dense infiltration with stromal elements, like fibroblasts, and myeloid cells, such as macrophages.

Fife et al. used multiphoton microscopy to test the hypothesis that PD-L1 prevented the T-cell-receptor-mediated stop signal associated with T cell activation [49]. PD-L1 blockade led T cells to slow down, indicating that PD-1–PD-L1 interactions prevent T cell stop signals. PD-L1 but not CTLA-4 blockade induced T cell arrest and limited displacement, suggesting that PD-L1 blockade enhances T cell stop signals. Importantly, anti-PD-L1 administration did not affect movement of naïve, polyclonal T cells outside the experimental setting.

In our hands, measurement of arrest coefficients showed an increase in the later observation phase after 18 h. This is concordant with deeper infiltration of T cells into the spheroid and more contact with infiltrating macrophages. This might also be caused by the dispersed presence of model antigen OVA in an already dissolving spheroid, due to cytotoxic efficacy of T cells. Concordantly, later observations demonstrated shrinking volume and an increase in spheroid fragmentation. LPS and nigericin treatment of macrophages decreased arrest phases of CTLs, indicating less cytotoxicity. All this suggests immunosuppressive, protumorigenic effects of intratumoral NLRP3 activation.

There are certain limitations to this study. Validation of NLRP3-related effects on T cell motility and effector function is difficult to achieve. We cannot prove that tracking arrests are, in fact, causally linked to cytotoxic effector function. However, demonstrating a causal relationship between visualization data and molecular mechanisms has always been problematic since the first reports on in vivo in situ imaging of T cells. Early reports could demonstrate that, e.g., a change in fluorescence properties of surrogate target cells is associated with target cell death [53], validating multiphoton imaging as a readout parameter of cytotoxic efficacy. Since then, migration pattern analysis has been clearly linked to effector function of CTLs [20]. We built on these previous reports when calculating arrest coefficients of CTL tracks as surrogate parameters of cytotoxicity. However, further studies are needed to bridge the gap between the visual impact of dynamic T cell migration in spheroids and the rather small (but consistent) molecular effects (of, e.g., NLRP3 activation in tumor macrophages).

An important caveat is that only one cell line (PancOVA) was used. However, two different conditions were investigated with T cells from two fully separated mouse lines, WT and *Il18r^−/−^* OT-I mice. Effects of LPS and nigericin treatment of macrophages could be reproduced in both settings, which was also mechanistically important, as they hint toward non-T-cell mechanisms. We speculate that this is mediated by multiple factors, among them increased expression of PD-L1, decreased expression of CD80 and CD86, and alteration of surface MHC-I expression. Further experiments including pharmacological pathway inhibition are warranted.

Beside insights into the molecular regulation of macrophage–T cell interaction, the novelty of this paper lies in establishing a triple-cell-type spheroid model that produces consistent motility measurements. Admittedly, differences generated by interventions such as addition of macrophages, IL-18R deficiency, or nigericin treatment are comparatively small and only partly significant. However, this paper demonstrates that motility alterations are paralleled by cytotoxicity and rejection kinetics of the tumor spheroids.

This study strengthens the notion that factors that influence T cell motility have a direct impact on T cell effector function. In a previous publication, our group demonstrated that T cell motility is highly concordant when measured in vivo in situ in a dorsal skinfold chamber model [8] versus a tumor spheroid model employing carcinoma cells and CTLs. Addition of macrophages not only as a passive “ingredient” of the spheroid seed but as an active contributor toward intratumoral motility has, to our knowledge, not been shown before. In general, T cell motility analysis is not a regular or often-used parameter of spheroid experiments. We propose that T cell migration analysis in tumor spheroids is a valuable tool to investigate T cell immunobiology.

Emphasis on T cell motility puts a different focus on cancer spheroids compared to other very recent reports on triple cell spheroid models of pancreatic cancer. He et al. used three-dimensional tumor cultures seeding human primary monocytes into the spheroid upon formation [54]. In this model, they investigated the T cell capacity to infiltrate and kill 3D tumor spheroids from male and female donors over a time course of 36 h in live imaging. Sarhan et al. investigated targeting of myeloid suppressive cells in order to investigate cytotoxic antitumor responses in pancreatic cancer [55]. Here, spheroid size and NK cell infiltration were assessed. Wehrli et al. employed patient-matched cancer-associated fibroblasts (CAFs) in patient-derived organoids [56]. Another recent report characterized tumor microenvironment remodeling in immunosuppressive tumor organoids [57]. None of these recent studies used live imaging for state-of-the-art motility analysis. Our paper intends to establish a firm connection between T cell migration and T cell function in spheroid models, calling for more sophisticated readouts in spheroid models.

Methods going beyond sheer size measurement and infiltration counts are also employed by other groups working on tumor spheroids. Concordant with our results on caspase-3/7 activation, performing caspase-3 activation analysis in time-lapse microscopy videos was used as an endpoint by Teijeira et al. recently [58]. Other groups are pushing even further. Sun et al. reported about a quadruple cell coculture spheroid model recently [59].

## 5. Conclusions

T cell motility analysis is feasible in a triple cell spheroid model of pancreatic cancer. Migration analysis in this model is robust enough to allow molecular interventions such as NLRP3 activation of seeded macrophages. Our data support the notion that T cell motility patterns indicate the functional state of effector T cells. We suggest that treatment of macrophages with LPS and nigericin, resulting in NLRP3 activation, induces an increase in T cell velocity, which is not mirrored by an increase in arrest coefficient and cytotoxic efficacy. Instead, activation of macrophages resulted in a decrease in T cell arrests on target cells and decreased spheroid rejection. Further studies are needed to delineate the complex interplay of immunosuppressive and immunoactivating effects of intratumoral NLRP3 activation.

## Figures and Tables

**Figure 1 cancers-17-00051-f001:**
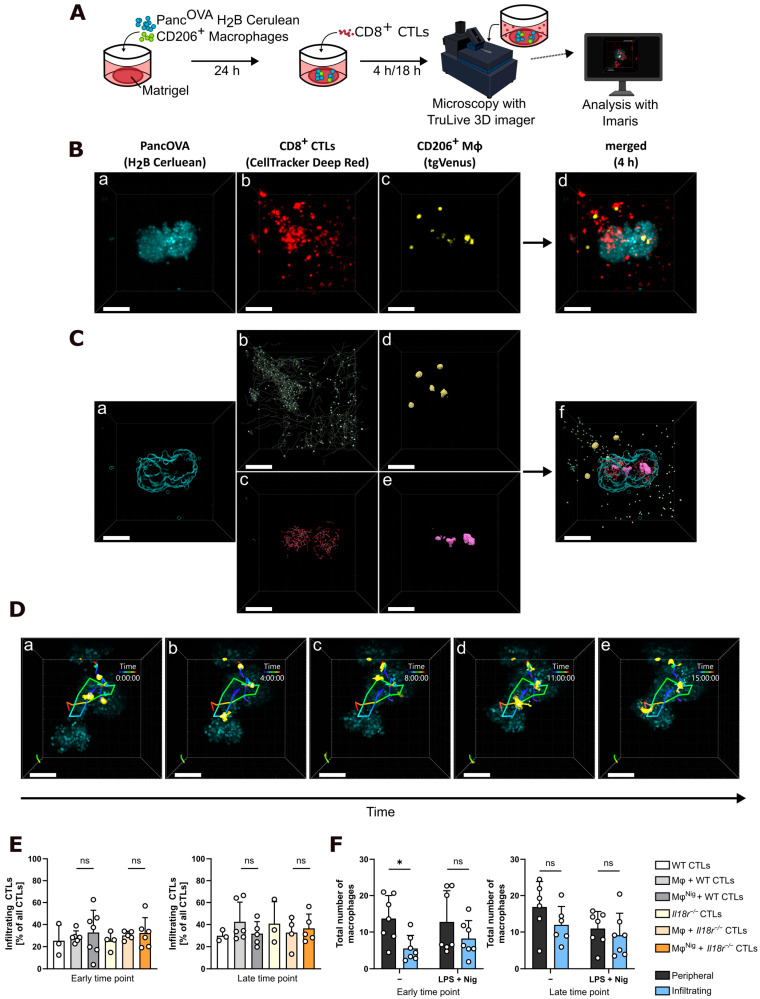
CD206^+^ macrophages infiltrate PancOVA spheroids and interact with OVA-reactive CD8^+^ CTLs. (**A**) Schematic experimental design. In total, 6 × 10^4^ with ovalbumin and fluorophore H2B Cerulean transfected Panc02 cells (PancOVA) were mixed in Matrigel with 8 × 10^3^ CD206^+^ macrophages. These were beforehand either not treated (Mφ) or treated with 200 ng/mL lipopolysaccharide (LPS) for 2 h and 10 µM nigericin for 1 h to activate the NLRP3 inflammasome (Mφ^Nig^). Cells were then transferred into TruLive 3D dishes. After 24 h of incubation, 1.2 × 10^5^ CD8^+^ CellTracker DeepRed-stained cytotoxic T cells (CTLs) were added to the dishes. After 4 h and 18 h, respectively, microscopic imaging was performed with the TruLive 3D Imager. (**B**) Representative three-dimensional (3D) spheroid experiment of tumor cells (turquoise, **a**), CD8^+^ CTLs (red, **b**), CD206^+^ macrophages (yellow, **c**), and all of the above (**d**) after 4 h of coculture. The white scale bar indicates 100 µm. (**C**) Representative tracking of tumor cells (turquoise, **a**), the peripheral population of CTLs (green, **b**), infiltrating CTLs (red, **c**), and peripheral (yellow, **d**), infiltrating (purple, **e**) CD206^+^ macrophages, and all of the above (**f**). The white scale bar indicates 100 µm. (**D**) Representative tracking of three tumor-infiltrating CD206^+^ macrophages (yellow) after 0 (**a**), 4 (**b**), 8 (**c**), 11 (**d**), and 15 (**e**) hours of coculture with time-color-coded tracks. The white scale bar indicates 80 µm. (**E**) Percentages of antigen-specific infiltrating CTLs of all CTLs in PancOVA spheroids after 4 h and 18 h of coculture using the spot algorithm of the Imaris software (*n* = 3–7). (**F**) Number of peripheral and spheroid-infiltrating CD206^+^ macrophages after 4 h and 18 h of coculture using the surface algorithm of the Imaris software (*n* = 6–7). -: no treatment with LPS and nigercin; LPS + Nig: activation of the NLRP3 inflammasome with 200 ng/mL LPS (2 h) and 10 µM nigericin (1 h). Statistical analysis was performed using a one-way ANOVA with Welch correction. All graphs are shown as mean ± standard deviation (SD). ns = not significant, * *p* ≤ 0.05.

**Figure 2 cancers-17-00051-f002:**
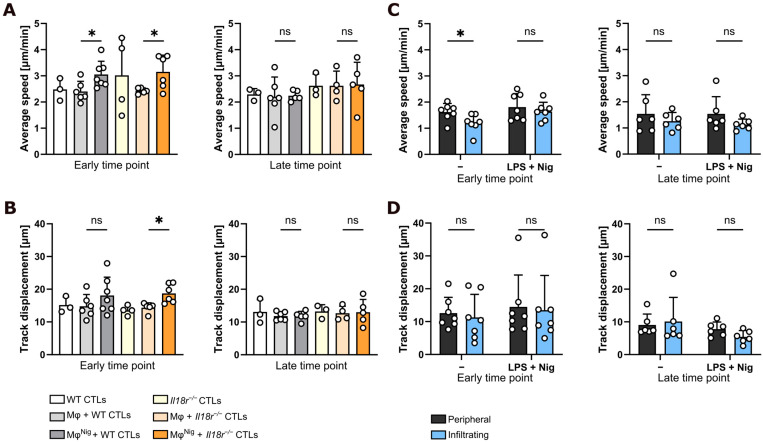
Motility of intratumoral CTLs is increased in the presence of CD206^+^ macrophages treated with LPS and nigericin. In total, 6 × 10^4^ PancOVA cells were mixed in Martigel with 8 × 10^3^ CD206^+^ macrophages, which were either not treated (Mφ) or treated with 200 ng/mL LPS (2 h) and 10 µM nigericin (1 h) to activate the NLRP3 inflammasome (Mφ^Nig^). After 24 h of incubation, 1.2 × 10^5^ CD8^+^ CellTracker DeepRed-stained CTLs were added to the dishes. Microscopic imaging was performed with the TruLive 3D Imager. A spot algorithm was used to track the migration pattern of wild-type (WT) and *Il18r^−/−^* CD8^+^ CTLs, and a surface algorithm was used to track the migration pattern of CD206^+^ macrophages in 30 min videos of PancOVA spheroid coculture at early (4 h) and late (18 h) time points. (**A**) Average speed of infiltrating WT and *Il18r^−/−^* CD8^+^ CTLs at the early and late time points (*n* = 3–7). (**B**) Track displacement of infiltrating WT and *Il18r^−/−^* CD8^+^ CTLs at early and late time points (*n* = 3–7). Track displacement is the vectorial distance from the start to the endpoint of a CTL track. (**C**) Average speed of infiltrating and peripheral CD206^+^ macrophages at early (4 h) and late (18 h) time points (*n* = 6–7). (**D**) Track displacement of infiltrating and peripheral CD206^+^ macrophages at early (4 h) and late (18 h) time points (*n* = 6–7). -: no treatment with LPS and nigericin; LPS + Nig: activation of the NLRP3 inflammasome with 200 ng/mL LPS (2 h) and 10 µM nigericin (1 h). Statistical analysis was performed using an unpaired, two-tailed *t*-test with Welch correction. All graphs are shown as mean ± SD. ns = not significant, * *p* ≤ 0.05.

**Figure 3 cancers-17-00051-f003:**
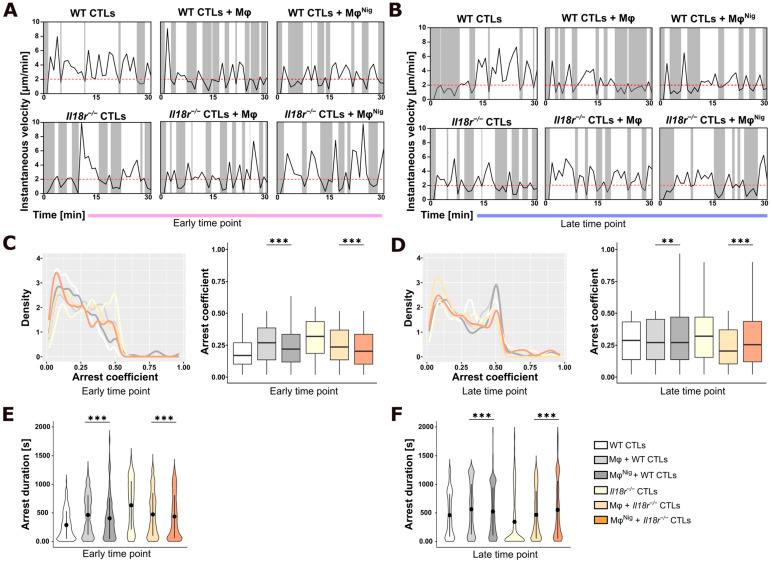
Interaction with CD206^+^ macrophages exerts epitope-specific T cell arrest in PancOVA spheroids. (**A**) Representative graphs of the instantaneous velocity of individual tracked infiltrating WT and *Il18r^−/−^* CD8^+^ CTLs in coculture with CD206^+^ macrophages and tumor spheroids at the early (4 h) time point. The graphs show the alteration of high and low instantaneous velocities. The threshold of instantaneous velocity of 2 µm/min is indicated as a dashed red line to mark the pausing phases. (**B**) Representative graphs of the instantaneous velocity of individual tracked infiltrating WT and *Il18r^−/−^* CD8^+^ CTLs at the late (18 h) time point. The threshold of instantaneous velocity of 2 µm/min is indicated as a dashed red line to mark the pausing phases. (**C**) Arrest coefficient and density of arrest coefficient of WT and *Il18r^−/−^* CD8^+^ CTLs cocultured with CD206^+^ macrophages in PancOVA spheroids at the early (4 h) time point. The arrest coefficient is the fraction of time an individual cell stays in arrest based on a threshold of 2 µm/min instantaneous velocity. The density plot depicts the distribution of arrest coefficients. Peaks describe the accumulation of arrest coefficients. The graphs represent pooled track data from every CTL track of 53 videos (*n* = 3–7). The arrest coefficients were calculated and plotted using RStudio. (**D**) Arrest coefficient and density of arrest coefficient at the late (18 h) time point. The graphs represent pooled track data from every CTL track of 36 videos (*n* = 3–7). The arrest coefficient was calculated and plotted using RStudio. (**E**) Duration of arrests of WT and *Il18r^−/−^* CD8^+^ CTLs cocultured with CD206^+^ macrophages in PancOVA spheroids after 4 h of coculture. The violin blots represent pooled track data from 53 videos (*n* = 3–7). The arrest durations were calculated and plotted using RStudio. (**F**) Duration of arrests of WT and *Il18r^−/−^* CD8^+^ CTLs after 18 h of coculture. The violin blots represent pooled track data from 36 videos (*n* = 3–7). The arrest durations were calculated and plotted using RStudio. Graphs and violin blot means are shown as confidence intervals (95% CI). Statistical analysis was performed using one-way ANOVA with Welch correction. * *p* ≤ 0.05, ** *p* < 0.01, and *** *p* < 0.001.

**Figure 4 cancers-17-00051-f004:**
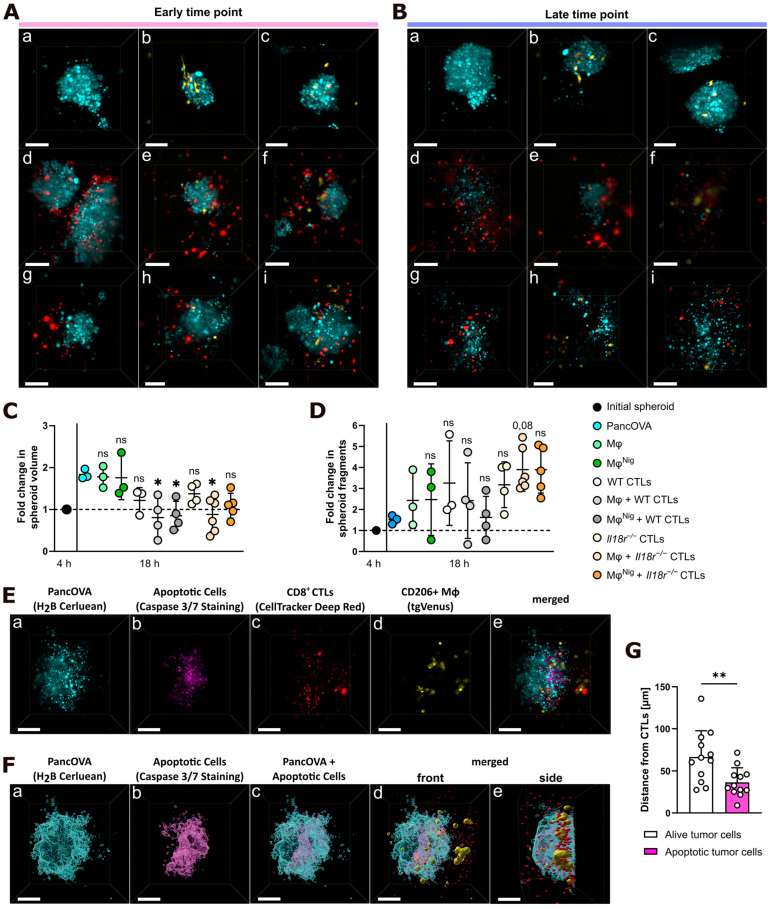
Interaction of T cells with macrophages in Matrigel PancOVA spheroids enhances CTL cytotoxicity. Representative 3D depictions of PancOVA spheroids alone or in different combinations of coculture after 4 h (**A**) or 18 h (**B**). Tumor cells are shown in turquoise, CD8^+^ CTLs in red, and CD206^+^ macrophages in yellow. Spheroids were cultured alone (**a**), with untreated macrophages (Mφ) (**b**,**e**,**h**), macrophages with activated NLRP3 inflammasome due to treatment with 200 ng/mL LPS (2 h) and 10 µM nigericin (1 h) (Mφ^Nig^) (**c**,**f**,**i**), WT CTLs (**d**–**f**), or *Il18r^−/−^* CTLs (**g**–**i**). Tumor size progression was observed overnight. The white scale bar indicates 100 µm. (**C**) Fold change in spheroid volume overnight. Each dot represents one experiment (*n* = 3–6). (**D**) Fold change in spheroid fragments overnight. Each dot represents one experiment (*n* = 3–6). (**E**) Representative three-dimensional depiction of PancOVA spheroids (turquoise, **a**), WT CD8^+^ CTLs (red, **c**) and CD206^+^ macrophages (yellow, **d**), and apoptotic PancOVA cells (purple, **b**) after 18 h of coculture. After 18 h of coculture, caspase-3/7 dye was added to the coculture system. The dye binds to the DNA when the cells enter induced cell death, producing a fluorescent reaction. (**e**) shows the merged staining. The white scale bar indicates 100 µm. (**F**) Representative algorithm masks for PancOVA spheroids (turquoise, **a**) and apoptotic PancOVA cells (pink, **b**) after 18 h of coculture with WT CD8^+^ CTLs (red) and CD206^+^ Mφ^Nig^ (yellow). After 18 h of coculture, Caspase 3/7 dye was added to the coculture system. (**c**–**e**) shows the distribution and localization of PancOVA (turquoise) and apoptotic cells (pink) alone (**c**) and with WT CD8^+^ CTLs (red) and CD206^+^ Mφ^Nig^ (yellow) (**d**,**e**). The white scale bar indicates 100 µm. (**G**) Distance from alive PancOVA cells or apoptotic PancOVA cells to CD8^+^ CTLs. Each dot represents the mean distance in one coculture experiment with WT CTLs or *Il18r^−/−^* CTLs (*n* = 12). The graph is shown as mean ± SD. Statistical analysis was performed using a one-way ANOVA followed by a Kruskal–Wallis test (**C**,**D**) or an unpaired, two-tailed *t*-test (**G**). ns = not significant, * *p* ≤ 0.05, ** *p* < 0.01.

## Data Availability

The original contributions presented in the study are included in the article/Appendix A; further inquiries can be directed to the corresponding author.

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
