# Peer review of "Optimized Spheroid Model of Pancreatic Cancer Demonstrates Influence of Macrophage–T Cell Interaction for Intratumoral T Cell Motility"

_cancers, 2024, doi:10.3390/cancers17010051_

Round 1
Reviewer 1 Report (Previous Reviewer 1)
Comments and Suggestions for Authors
The authors have made amendments to the manuscript pointing to the importance of the novel spheroid model being described. The model itself is a useful methodology to study interactions in the tumor microenvironment. However, some of the concerns still remain. The authors should remove mentioning in the text and conclusion of the manuscript that the cause of the observed effect is NLRP3 inflammasome activation. Instead, they can stick to the facts by saying that Nigercin treatment induced the observed effects. Since Nigercin has many targets including NLRP3 and the experiments haven't exclusively confirmed the role of NLRP3 it will be factually incorrect to make such statements. For eg: headings like "NLRP3 Activation Abrogates Macrophage-induced Increase of T Cell Arrests in PancOVA 366 Spheroids"should be avoided. Instead, it can be mentioned that "Nigercin treatment Abrogates Macrophage-induced Increase of T Cell Arrests in PancOVA 366 Spheroids". The authors can discuss the possibility of this being driven through NLRP3 activation in the discussion part or results.
Author Response
The authors have made amendments to the manuscript pointing to the importance of the novel spheroid model being described. The model itself is a useful methodology to study interactions in the tumor microenvironment. However, some of the concerns still remain. The authors should remove mentioning in the text and conclusion of the manuscript that the cause of the observed effect is NLRP3 inflammasome activation. Instead, they can stick to the facts by saying that Nigercin treatment induced the observed effects. Since Nigercin has many targets including NLRP3 and the experiments haven't exclusively confirmed the role of NLRP3 it will be factually incorrect to make such statements. For eg: headings like "NLRP3 Activation Abrogates Macrophage-induced Increase of T Cell Arrests in PancOVA 366 Spheroids"should be avoided. Instead, it can be mentioned that "Nigercin treatment Abrogates Macrophage-induced Increase of T Cell Arrests in PancOVA 366 Spheroids". The authors can discuss the possibility of this being driven through NLRP3 activation in the discussion part or results.
Thank you for commenting favorably on our model and the methodological advance. We agree that we did not formally demonstrate NLRP3 activation through LPS and nigericin treatment, nor did we rule out off-target effects of nigericin. Therefore, we understand that the reviewer suggests to limit ourselves to the technical aspect of macrophage activation through LPS and nigericin treatment, leaving the exact mechanisms as part of the discussion.
We agree that our findings on T cell motility in a three-cell-type-spheroid model are valuable even without formal proof of NLRP3 activation, as the manuscript would work well by just describing CTL motility effects after activating macrophages or not (leaving the mechanism of LPS/nigericin treatment completely unmentioned). However, we believe that strictly cancelling the aspect of assumed NLRP3 activation from the text (minus a short paragraph on possible NLRP3 activation in the discussion) would make it very hard for the reader to understand why we opted for LPS/nigericin pretreatment of macrophages.
In our hands, nigericin resulted reliably in NLRP3 activation, formally demonstrated in earlier work from our group (including western blotting and speck formation assays; Bauer et al. Gut 2010). As demonstrated in our earlier rebuttal letter after the first submission of this paper, LPS and nigericin treatment of macrophages resulted in IL-1beta and IL-18 release.
We have rephrased all segments of the text in a more cautious way, sticking to the technical aspect of LPS and nigericin treatment, and mentioning NLRP3 activation as an intention of this treatment, not as a fact. A paragraph on the mechanisms of nigericin-induced NLRP3 activation and other targets of nigericin has been included into the manuscript (lines 446 to 464).
Reviewer 2 Report (Previous Reviewer 2)
Comments and Suggestions for Authors
This is a manuscript that was once reviewed by me and I supported publishing it with minor revision. Now the same manuscript was resubmitted, titled “Optimized Spheroid Model of Pancreatic Cancer demonstrates Influence of Macrophage-T Cell Interaction for intratumoral T Cell Motility”. It describes their findings on the role of NLRP3 activation in myeloid cells on CTL motility in a refined spheroid model of pancreatic cancer. This study tried to establish a triple cell spheroid model of pancreatic carcinoma that characterizes intratumoral T cell motility as a parameter of T cell effector function. The authors investigated primarily the motility analysis of macrophages and changes in motility of CTLs as a result of CTL–macrophage interaction. Pancreatic cancer is a major concern and the extensive research in this area is highly appreciated. It is interesting to see the spheroid model approach, a little different approach than conventional methods. It is exciting to see the outcome of their work and potential future research plan. The manuscript is well written and I am satisfied with the current data and analysis.
I do not have any major concerns regarding the current version of the submitted manuscript. With the minor changes needed after proofreading, I support publishing this manuscript in ‘cancers’.
Author Response
This is a manuscript that was once reviewed by me and I supported publishing it with minor revision. Now the same manuscript was resubmitted, titled “Optimized Spheroid Model of Pancreatic Cancer demonstrates Influence of Macrophage-T Cell Interaction for intratumoral T Cell Motility”. It describes their findings on the role of NLRP3 activation in myeloid cells on CTL motility in a refined spheroid model of pancreatic cancer. This study tried to establish a triple cell spheroid model of pancreatic carcinoma that characterizes intratumoral T cell motility as a parameter of T cell effector function. The authors investigated primarily the motility analysis of macrophages and changes in motility of CTLs as a result of CTL–macrophage interaction. Pancreatic cancer is a major concern and the extensive research in this area is highly appreciated. It is interesting to see the spheroid model approach, a little different approach than conventional methods. It is exciting to see the outcome of their work and potential future research plan. The manuscript is well written and I am satisfied with the current data and analysis.
I do not have any major concerns regarding the current version of the submitted manuscript. With the minor changes needed after proofreading, I support publishing this manuscript in ‘cancers’.
We thank the reviewer for the favorable recommendation.
This manuscript is a resubmission of an earlier submission. The following is a list of the peer review reports and author responses from that submission.
Round 1
Reviewer 1 Report
Comments and Suggestions for Authors
The authors have developed an interesting model to study interactions between tumor cells, M2-polarized macrophages, and CD8 T cells. However, the validity of the model in studying these interactions remains a question. Also, the conclusions drawn from the model are not clear. It is unclear how the addition of M2 macrophages is affecting T cell biology in this model. There is little data clearly showing how the addition or activation of macrophages is affecting T cell biology ( activation/ exhaustion etc.) or its functionality. The data provided shows that the effects of NLRP3 activation using Nigericin on T cell function are weak. While NLRP3 activation seems to increase T cell motility, it appears to be negatively regulating T cell cytotoxicity. The authors should test for markers showing activation/ exhaustion and cytotoxicity of T cells in co-culture. It is unclear from the data whether the IL18 receptor plays any significant role in this model. The authors also do not show any direct confirmation of NLRP3 activation in macrophages and if activated, it is unclear whether the macrophages still have active NLRP3 while in coculture. Neither there is confirmation of increased cytokine release like IL1 Beta or IL18. The authors also should use an NLRP3 inhibitor to test whether the observed effects are through NLRP3 as LPS and nigericin also have other targets and effects. It is also unclear why the last figure does not compare Caspase 3 and 7 activations across conditions. Together although the model seems interesting much deeper characterization of the model is required before drawing any valid conclusions.
Reviewer 2 Report
Comments and Suggestions for Authors
Pancreatic cancer is a major concern and the extensive research in this area is highly appreciated especially combination therapies. It is interesting to see the spheroid model approach, a little different approach. In the manuscript “Intratumoral T Cell Migration Is Enhanced by NLRP3-dependent Macrophage–T Cell Interaction in a Pancreatic Cancer Spheroid Model” by Slusny et.al investigated the role of NLRP3 activation in myeloid cells on CTL motility in a refined spheroid model of pancreatic cancer. It is interesting to see the outcome of the presented work and potential future research plan. I am satisfied with the current data and the manuscript is well written.
The style of the abstract is boring and there is no need to classify it as introduction, methods etc. The flow of the abstract is not appealing and I recommend rewriting it.
Upon incorporating the minor changes suggested, I support publishing this manuscript in ‘cancers’.
Reviewer 3 Report
Comments and Suggestions for Authors
The manuscript " Intratumoral T Cell Migration Is Enhanced by NLRP3-dependent Macrophage–T Cell Interaction in a Pancreatic Cancer Spheroid Model” by Slusny et al is an interesting research article. This is an impressive body of work, rich in data and significance. However, a major concern is the author's decision not to include human pancreatic cell lines (clinical perspective). Is there a specific reason for focusing solely on mouse-derived cell lines for this study? Additionally, the entire interpretation and hypothesis appear to be validated using only a single cell line, which raises questions about the robustness of the findings. There are several missing details in the manuscript that need to be addressed, as described below.
1. There is no need to include references in the abstract. This information can be incorporated into the Introduction with appropriate citations.
2. The authors should include data that supports the identity and authentication of the cell line used in the manuscript.
3. For the animal work, please provide the number of mice used and any relevant approval details.
4. The Discussion should be significantly condensed. As it's written, the Discussion appears to mostly recap their results and lacks a critical evaluation of their results.
Round 2
Reviewer 1 Report
Comments and Suggestions for Authors
The explanations provided by the Authors are just pointing to some of the previous publication findings. They do not really explain the validity of this particular model, especially when looking at the role of M2 macrophages. It is not clear whether the study can be used to draw any meaningful conclusions on the role of macrophages in T cell motility or cytotoxicity. It is also not clear whether the presence or absence of IL18 receptor has any additional effect on the influence of M2 macrophage on T cell motility or cytotoxicity. Even if there are minor differences in these parameters there is no clear explanation/ reasoning given by the authors to understand its relevance in this context. Also, the authors cannot claim the effect is through NLRP3 activation when there are no experiments performed to clearly delineate the role of NLRP3 activation. Correlation cannot be used to claim causation. Together the study does not provide any meaningful novel conclusions on the biology of T cell and M2 macrophage interaction. The 3d model described is quite robust and could be useful for future studies.
Reviewer 3 Report
Comments and Suggestions for Authors
All comments have been appropriately addressed by the authors. Furthermore, the authors have provided valid justifications for not including certain suggested experiments, which can be considered acceptable.
Thanks!